# Nucleolar Stress Functions Upstream to Stimulate Expression of Autophagy Regulators

**DOI:** 10.3390/cancers13246220

**Published:** 2021-12-10

**Authors:** David P. Dannheisig, Anna Schimansky, Cornelia Donow, Astrid S. Pfister

**Affiliations:** Institute of Biochemistry and Molecular Biology, Ulm University, Albert-Einstein-Allee 11, D-89081 Ulm, Germany; david.dannheisig@uni-ulm.de (D.P.D.); annaschimansky@googlemail.com (A.S.); cornelia.donow@uni-ulm.de (C.D.)

**Keywords:** autophagy, ATG7, ATG16L1, nucleolus, nucleolar stress, CX-5461, PPAN, NPM, SBDS, PES1, UBF-1, ribosome biogenesis

## Abstract

**Simple Summary:**

Ribosome biogenesis takes place in nucleoli and is essential for cellular survival and proliferation. In case this function is disturbed, either due to defects in regulatory factors or the structure of the nucleolus, nucleolar stress is provoked. Consequently, cells classically undergo cell cycle arrest and apoptosis. Induction of nucleolar stress is known to eliminate cells in the background of cancer therapy and paradoxically is also associated with increased cancer formation. Recent reports demonstrated that nucleolar stress triggers autophagy, a conserved pathway responsible for recycling endogenous material. Thus, it was suggested that autophagy might serve as compensatory pro-survival response. However, the mechanisms how nucleolar stress triggers autophagy are poorly understood. Here we show that induction of nucleolar stress by depleting ribosome biogenesis factors or by interfering with RNA polymerase I function, triggers expression of various key autophagy regulators. Moreover, we demonstrate that RNA pol I inhibition by CX-5461 correlates with increased *ATG7* and *ATGL16L1* levels, essential factors for generating autophagosomes, and stimulates autophagic flux.

**Abstract:**

Ribosome biogenesis is essential for protein synthesis, cell growth and survival. The process takes places in nucleoli and is orchestrated by various proteins, among them RNA polymerases I–III as well as ribosome biogenesis factors. Perturbation of ribosome biogenesis activates the nucleolar stress response, which classically triggers cell cycle arrest and apoptosis. Nucleolar stress is utilized in modern anti-cancer therapies, however, also contributes to the development of various pathologies, including cancer. Growing evidence suggests that nucleolar stress stimulates compensatory cascades, for instance bulk autophagy. However, underlying mechanisms are poorly understood. Here we demonstrate that induction of nucleolar stress activates expression of key autophagic regulators such as *ATG7* and *ATG16L1*, essential for generation of autophagosomes. We show that knockdown of the ribosomopathy factor SBDS, or of key ribosome biogenesis factors (PPAN, NPM, PES1) is associated with enhanced levels of ATG7 in cancer cells. The same holds true when interfering with RNA polymerase I function by either pharmacological inhibition (CX-5461) or depletion of the transcription factor UBF-1. Moreover, we demonstrate that RNA pol I inhibition by CX-5461 stimulates autophagic flux. Together, our data establish that nucleolar stress affects transcriptional regulation of autophagy. Given the contribution of both axes in propagation or cure of cancer, our data uncover a connection that might be targeted in future.

## 1. Introduction

Ribosome biogenesis is essential for growth and survival of cells. It occurs in sub-nuclear compartments, the nucleoli, which accommodate a plethora of processing factors to mediate maturation of rRNA [1]. The dynamic process involves transcription of the *47S rRNA* precursor by RNA polymerase I (RNA pol I), followed by cleavage and maturation of the transcript by processing factors and assembly of mature rRNAs with ribosomal proteins to form the two ribosomal subunits [2]. Moreover, the nucleolus functions as critical cellular stress sensor and central hub in the nucleolar stress response. The term nucleolar stress denotes a key response to non-functional ribosome biogenesis or nucleolar disruption. Triggers can either be cellular or drug-induced stress that interfere with nucleolar structure or integrity, as well as loss or mutation of factors implicated in ribosome biogenesis. Nucleolar stress classically results in stabilization of the tumor suppressor and “guardian of the genome” p53, which in turn mediates cell cycle arrest, senescence or apoptosis [3].

Meanwhile, inducing nucleolar stress-mediated apoptosis has gained tremendous interest for developing modern anti-cancer strategies. Thereby, already established chemotherapeutics such as ActinomycinD, Methotrexate or 5-Fluorouracil interfere with nucleolar function and aim to trigger apoptotic cell death of cancer cells [4]. The small-molecule drug of the newer generation, CX-5461, specifically interferes with RNA pol I activity and thus triggers nucleolar stress, without affecting RNA pol II function [5]. 

Note that nucleolar stress can also stimulate p53-independent, pro-apoptotic responses [6,7]. These mechanisms are of particular relevance for anti-tumor therapies [8], given the common inactivation of the tumor suppressor p53 in diverse cancers [9]. Besides its impact on apoptosis and cell cycle progression, as previously demonstrated in aggressive acute myeloid leukemia (AML), CX-5461 has been shown to induce senescence and stimulate autophagy in solid tumors independently of functional p53 [10,11]. Thus, CX-5461 possesses multiple aspects, which contribute to its anti-tumor activity, previously demonstrated in phase I clinical trials [12].

Strikingly, induction of nucleolar stress is not always connected to cell death but can have even counter-intuitive effects such as cancer formation. In so-called ribosomopathy syndromes, which are characterized by haploinsufficiency of ribosome biogenesis factors or ribosomal proteins, an increased cancer incidence is observed [13,14]. This reflects an unsolved paradoxon, termed “Dameshek’s riddle”, given the fact that hyper-proliferating cells possess a high demand of ribosomes [15]. Ribosomopathy patients commonly develop AML as well as diverse solid tumor types such as cervical cancer, osteosarcoma or colon cancer [16,17,18]. Null mutations in ribosomopathy syndromes are not found, given the fact that a complete loss is embryonically lethal due to a reduced ribosome number [15]. Examples include Shwachman-Bodian-Diamond-Syndrome (SBDS) in which the SBDS protein is mutated [19]. 

In addition, mutation of the nucleolar factor Nucleophosmin (NPM) is frequently found in AML and aberrant expression of the ribosome biogenesis factors Peter Pan (PPAN) or Pescadillo (PES1) have been connected to tumorigenesis [15,20,21,22,23]. Importantly, a constantly increasing repertoire of responses is being uncovered as consequence of nucleolar malfunction that might account at least for some of the observed phenotypes. For instance, nucleolar stress has recently been identified as upstream trigger of certain signaling cascades, such as the pro-proliferative Wnt/β-Catenin pathway or autophagy [15,24,25]. However, further research is essential to decipher the cellular mechanisms underlying the consequences of nucleolar stress induction.

Autophagy is stimulated by various types of stress, for instance nutrient deprivation, lack of energy—and as recently found—also nucleolar stress. Macro-autophagy, commonly referred to as autophagy, is a highly conserved and essential mechanism that orchestrates degradation or recycling of cellular material. Key to autophagy is the formation of double-membranous autophagosomes, which function in the sequestration and engulfment of material destined for degradation [26]. Autophagosomes are built with the help of autophagy related proteins (ATGs). ATG7, for instance, functions as an essential initiation factor and E1-like enzyme required for assembly of the autophagosomal membrane [27,28,29]. Together with other ATGs, ATG7 facilitates lipidation of Atg8/LC3. Also, ATG16L1 is crucial for autophagosome biogenesis and lipidation of LC3. This modified version, termed LC3-II, gets incorporated into the autophagosomal membrane through addition of its phosphaditylethanolamine anchor [30]. This step is essential for establishment, maturation and expansion of the autophagosomal membrane. The degradation of the material is in turn mediated by acidic lysosomes, which fuse with autophagosomes containing the engulfed cargo. 

Besides nucleolar stress induction by CX-5461, the chemotherapeutic ActinomycinD also has been shown to stimulate autophagy in cancer cells [24,31,32]. Moreover, loss of key nucleolar ribosome biogenesis factors connected to nucleolar stress induction, such as PPAN, NPM or the ribosomopathy factor SBDS have been linked to autophagy [15,31,33,34]. Overall, the increase of autophagic flux in the context of nucleolar stress is believed to initially possess compensatory anti-apoptotic effects. However, after continuous stress cells can still commit apoptosis at a point of no return [24]. This is in line with the fact that similar upstream mechanisms can trigger both, apoptosis and autophagy [35]. 

Given the neccessity of cellular clearance for sustaining cellular homeostasis, it is quite obvious that autophagy is connected to diverse pathological conditions. Autophagy has been extensively studied in cancer and was considered to play a dual role in tumorigenesis, depending on the context. On the one hand it is protective for the integrity of tumor cells, but on the other hand also prevents against tumor initiation: Whereas a stress-induced increase in autophagy is noticed as mechanism of drug-resistance in e.g. AML and against chemotherapy-induced apoptosis, inhibition of autophagy emerges as attractive mechanism for fighting cancer [36]. Note that inhibition of autophagy early in the development of cancer formation and tumor initiation can even promote tumorigenesis. At later stages tumors become more dependent on the process of autophagy, for instance in response to chemotherapy treatment [36]. The use of hydroxy-chloroquine, an inhibitor of lysosomal function and autophagy, has shown benefit in anti-cancer therapy by counteracting drug-resistance, and in particular a combined therapy together with apoptosis inducers resulted in robust cell death of cancer cells in clinical studies [37]. 

Whereas the rather novel link of nucleoli and bulk autophagy is emerging, the underlying causes remain largely elusive and require more mechanistical investigation. Here, we demonstrate that induction of nucleolar stress by multiple mechanisms, such as treating cancer cells with the RNA pol I inhibitor CX-5461 or interfering with the transcription factor of RNA pol I, Upstream-binding factor 1 (UBF-1; UBTF), induces expression of the key regulator of autophagy, ATG7. Moreover, the effect can be reproduced by depletion of PPAN, SBDS, NPM and PES1 thereby strongly suggesting a common mechanism of action. Thus, our data establish the nucleolar stress response as upstream trigger for transcriptional regulation of autophagy. Given the contribution of both axes, nucleolar stress and autophagy, in propagation or cure of cancer our data unravel a link that might be a feasible target for future therapies.

## 2. Materials and Methods

### 2.1. siRNA Sequences

The following siRNAs were Flexitube siRNAs from Qiagen (Hilden, Germany): si PPAN-A (SI00125545) [25,34,38,39], si SBDS-C (SI00711144) and si SBDS-D (SI03246390) [25], si UBF-A (SI00754992) and si UBF-B (SI04290839). Following custom si RNA sequences were previously reported [25,34,38,39] and were obtained from Horizon Discovery (Lafayette, CO, USA). The respective si RNA sequences are listed below:

si control: 5′-GCUACCUGUUCCAUGGCCA-3′

si PPAN-B: 5′-GGACGAUGAUGAACAGGAA-3′

si NPM-A: 5′-UGAUGAAAAUGAGCACCAG-3′

si NPM-B: 5′-GAAUUGCUUCCGGAUGACU-3′

si PES1: 5′-CCAGAGGACCUAAGUGUGA-3′

### 2.2. Antibodies and Dyes

Commercial antibodies were obtained from following companies: Abnova (Taipei City, Taiwan): UBTF (H00007343-M01), Cell Signaling (Danvers, MA, USA): GAPDH (1410C), p21 (2946); Proteintech (Rosemont, CA, USA): PPAN (11006-1-AP), ATG7 (67341-1-Ig) and SBDS (67200-1-Ig); Sigma (Steinheim, Germany): PES1 (SAB1400457); Thermo Fisher Scientific (Dreieich, Germany): NPM (FC-61991); Roche (Mannheim, Germany): GFP (11814460001); Santa Cruz (Dallas, TX, USA): p53 (sc-126). Secondary antibodies for Western blotting were IRDye conjugates 800CW and 680CW from Li-COR (Lincoln, NE, USA) or Alexa Fluor 594 (Dianova, Hamburg, Germany) for immunofluorescence experiments. DAPI mounting medium were purchased from Dianova (Hamburg, Germany). 

### 2.3. Cell Culture, Transfections and Drug Treatments

Cells were grown in DMEM (high glucose) supplemented with penicillin and streptomycin, 10% FCS and were cultured at 37 °C and 5% CO_2_. U2OS cells were purchased from ATCC and HeLa cells were kindly provided by M. Kühl and were purchased from Sigma (Steinheim, Germany). HEK293A GFP-LC3 cells (ECACC 14050801) were purchased from Sigma (Steinheim, Germany) and were cultured in medium supplemented with 0.4 mg/mL G418 (Sigma, Steinheim, Germany) [34,40]. Cell lines were regularly tested for absence of mycoplasma and were used in low passage numbers.

HeLa and U2OS cells were seeded in 6-wells and were transfected with 45 nM si RNAs for 48 h using Oligofectamine according to the manufacturer´s instructions. For drug treatment, cells were seeded in 6-wells and grown over night prior administration of DMSO (Sigma) or 1 µM CX-5461 (Caymen, Ann Arbor, MI, USA) for 24 h as indicated. For flux assays in HEK293A GFP-LC3, cells were incubated with DMSO or 1 µM CX-5461 for 20.5 h followed by 50 µM chloroquine (CQ; Sigma, Steinheim, Germany) for 3.5 h. 

### 2.4. Cell Lysates, Western Blots and Densitometry

Cells were rinsed in 1× PBS, afterwards lysed with RIPA buffer (50 mM Tris-HCl pH 8.0, 150 mM NaCl, 1% NP-40, 0.1% SDS, 0.5% sodium deoxycholate) for 10 min on ice and cleared by centrifugation for 10 min at 13,000 rpm and 4 °C as previously [25]. 

Bradford assay, SDS-PAGE and Western blotting were performed using the Bio-Rad system. Nitrocellulose membranes were incubated over night with primary antibodies (dilutions 1:500 to 1:3000) followed by secondary antibody incubation for 2 h at room temperature (1:10,000). Protein levels were quantified using a Li-COR CLX Imager and Image Studio Light software as previously described [25]. For normalization, respective GAPDH signals of the same sample and blot were used. 

### 2.5. qRT-PCR Analysis

Total RNA was isolated using the RNeasy Mini Kit provided by Qiagen (Hilden, Germany). RNA samples were treated with DNAse (Roche, Mannheim, Germany) prior reverse transcription into cDNA using random hexamer primers and SuperScriptII (Invitrogen, Carlsbad, CA, USA). Thereby, samples without reverse transcriptase served as a contamination control. For qRT-PCR analysis, the QuantiTect SYBR Green PCR Kit (Qiagen, Hilden, Germany) was used with the CFX Connect Real Time System (BioRad, Hercules, CA, USA). Each experiment was measured in technical triplicates and was performed at least four times. The relative expression R was calculated according to the efficiency corrected method for HeLa cells as previously [25]. For U2OS, non-transfected HeLa as well as HEK293A GFP-LC3 cells an efficiency of E = 2 was assumed. The *PPAN*, *NPM1*, *PES1*, *SBDS*, *CDKN1A* and *47S rRNA* qRT-PCR primers were synthesized by biomers.net GmbH (Ulm, Germany) and were as previously reported [25,39]. The qRT-PCR primer sequences newly utilized in the present study were:

ATG4A-fw: 5′-CCTTCAGTTGCATTGGGATT-3′

ATG4A-rv: 5′-TTCCTTCTGAACAAGGCTACAC-3′

ATG5-fw: 5′-GGCCATCAATCGGAAACTC-3′

ATG5-rv: 5′-GGTCTTTCAGTCGTTGTCTG-3′

ATG7-fw: 5′-CATGAGTTGACCCAGAAGAAG-3′

ATG7-rv: 5′-CAGCAGAGTCACCATTGTAG-3′

ATG9A-fw: 5′-CACCGGCTTATCAAGTTCATC-3′

ATG9A-rv: 5′-ATACGGAAGGGCAGACATAG-3′

ATG16L1-fw: 5′-GTTTCTGGGACATTCGATCAG-3′

ATG16L1-rv: 5′-CTCAGTCCTTTCTGGGTTTAAG-3′

UBTF-fw: 5′-GACCGTGCAGCATATAAAG-3′

UBTF-rv: 5′-ACTTGGACTGCAGAGTAG-3′

ULK1-fw: 5′-CTGGTTATGGAGTACTGCAAC-3′

ULK1-rv: 5′-AGGAAGAGCCTGATGGTGTC-3′

Resulting *ATG4A*, *ATG5*, *ATG7*, *ATG9A*, *ATG16L1*, *UBTF* and *ULK1* qRT-PCR amplicons were verified by gel electrophoresis as well as sequencing (Eurofins Genomics, Ebersberg, Germany).

For screening of alterations concerning autophagy related genes, si control or si PPAN-B cDNA samples derived from HeLa cells were initially tested for PPAN knockdown and subsequently subjected to the predesigned 96-well panel Autophagy (SAB Target List) H96 (BioRad, Hercules, CA, USA) according to the manufacturer´s protocol. The expression of target genes was normalized to the four stable reference genes *ACTB*, *GAPDH*, *HPRT1* and *TBP*. The respective runs passed all internal controls as depicted in the following Table 1.

### 2.6. Immunofluorescence

HEK293A GFP-LC3 cells were grown on coverslips overnight and treated with drugs as indicated. Cells were fixed with ice-cold methanol for 10 min at −20 °C and subsequently permeabilized with 0.5% Triton X-100 for 15 min at RT. Cells were blocked in 0.5% BSA (in 1× PBS) for 45 min at RT, incubated with GAPDH antibody (1:100) for 2 h at RT followed by incubation with Alexa Fluor 594 (1:1000) for 1 h at RT and finally mounted in DAPI mounting medium. 

### 2.7. Image Acquisition and GFP-LC3 Assay

GFP-LC3 puncta analysis was performed as previously described [34]. Shortly, z-stack images were taken with the Zeiss Axio Observer 7 microscope using the 63× objective with the Apotome.2 function. Slices in the z-plane were taken with an interval of 0.24 µm through the entire cell monolayer. Five images per condition were taken in random optical fields, thereby imaging approximately 50 cells per *n*. Subsequently, the z-stack images were processed with the z project module from ImageJ using the Max intensity projection type, providing a composite image of all planes. For quantitative analysis CellProfiler^TM^ (version 3.1.9) was used [41,42]. Samples of the same assay were imaged and processed equally.

### 2.8. Statistical Analysis

Statistical analysis was performed with GraphPad Prism software version 9 using at least four independent experiments (*n*) as described previously [25,34,39]. Statistical analysis of relative data such as Western blotting or qRT-PCR experiments was calculated using the one-tailed paired *t*-test and a normal distribution was assumed as reported [43]. GFP-LC3 puncta experiments were calculated by unpaired *t*-test. Normalization was performed for si control or DMSO treated cells as indicated in the figure legends. Error bars indicate S.D., statistically significant differences are indicated by asterisks.

## 3. Results

### 3.1. Depletion of the Key Ribosome Biogenesis Factor PPAN Stimulates Transcription of ATG7 and ATG16L1 and Results in Increased Protein Levels of ATG7

We previously found that knockdown of the ribosome biogenesis factor PPAN induced nucleolar stress, which triggered cell cycle arrest and apoptosis [38,39]. Moreover, PPAN knockdown activated autophagic flux in cancer cells [34]. To elucidate a feasible molecular mechanism possibly underlying activation of autophagy, a commercially available qRT-PCR screen for autophagy-related genes was initially performed. Besides other targets, siRNA mediated knockdown of PPAN resulted in upregulation of the autophagy factor *ATG7* in HeLa cervical cancer cells when using the functional PPAN si RNA, termed si PPAN-B (Figure 1, Appendix A) as previously [34,38]. Note that HeLa cells are considered a p53-negative cell line, which is important with respect to nucleolar stress induction in response to PPAN depletion [38].

Given the key role of ATG7 in the process of autophagy, we aimed to verify these initial findings in more detail. In addition, we also determined levels of other hits of the screen, such as *ATG16L1*. Indeed, the effect on both factors could be reproduced by two independent and functional PPAN siRNAs (si PPAN-A and -B) [25,34,38,39] in the cervical cancer cell line HeLa (Figure 2A–C). Both siRNAs have earlier been used to demonstrate effects of PPAN depletion on activation of autophagic flux [34]. As a basis for its functional role in autophagy, we next determined whether ATG7 is stabilized on protein level. To address this question, we performed quantitative Western blotting and analyzed protein levels of ATG7 in HeLa cells upon knockdown of PPAN. In compliance with the mRNA data, we found significantly increased ATG7 protein levels after PPAN knockdown by two independent siRNAs (Figure 2D,E). 

### 3.2. Depletion of the Key Ribosome Biogenesis Factors NPM, SBDS or PES1 and the Transcription Factor UBF-1 Stimulates Transcription of ATG7 and ATG16L1

Given the emerging role of nucleolar stress in autophagy, we investigated if knockdown of additional nucleolar factors such as NPM, SBDS or PES1, well known to trigger the nucleolar stress response, can recapitulate these findings. Moreover, we analyzed knockdown of Upstream-binding factor 1 (UBF-1, UBTF), the transcription factor of RNA pol I to investigate if the effect on *ATG7* may represent a general result of nucleolar stress induction.

As previously, two siRNAs each were used for NPM and SBDS depletion, while PES1 knockdown was determined with one earlier published functional siRNA [25,44]. For UBF-1 we tested two siRNAs in this study. Knockdown of all factors was verified by qRT-PCR in HeLa cells (Figure 3A,D,G,J). At the same time, each knockdown condition was accompanied by significantly increased *ATG7* levels in HeLa, with exception of one UBF siRNA (Figure 3B,E,H,K). In addition, also *ATG16L1* levels were significantly increased in all knockdown cells (Figure 3C,F,I,L). 

Moreover, we measured mRNA levels of additional autophagy regulators and targets of the initial autophagy screen panel (compare Figure 1), such as *ATG4A*, *ATG5*, *ATG9A* and *ULK1* after PPAN, NPM, SBDS, UBF-1 and PES1 knockdown in Hela cells (Appendix A). Overall, each knockdown condition showed increased levels of further candidates. SBDS knockdown even showed increased levels of all analyzed genes (Appendix A).

*ATG7* induction was also determined in the p53-positive osteosarcoma cell line U2OS and was chosen as an independent cellular model previously connected to autophagy in response to nucleolar stress [24,32,34]. Indeed, the effects on *ATG7* were also apparent in U2OS cells after depletion of nucleolar factors PPAN, NPM, SBDS, UBF-1 or PES1 (Figure 4). Together, this suggested that the observed transcriptional elevation of *ATG7* mRNA may represent a general mechanism of nucleolar stress induction.

### 3.3. Depletion of the Key Ribosome Biogenesis Factors NPM, SBDS, PES1 or UBF-1 Results in Increased Protein Levels of ATG7

NPM and SBDS have earlier also been linked to autophagy [24,31,33]. Thus, we next determined, whether ATG7 is stabilized on protein level, similar as observed after PPAN depletion. To do so, we analyzed protein levels of ATG7 in HeLa cells upon knockdown of NPM and SBDS. In addition, we tested knockdown of UBF-1 and PES1. Correlating with the qRT-PCR data, we found significantly increased ATG7 protein levels after NPM knockdown by two independent siRNAs (Figure 5A–C). The same was true after knockdown of SBDS (Figure 5D–F), UBF-1 (Figure 5G–I) or PES1 (Figure 5J–L). Taken together depletion of key nucleolar factors, well associated with nucleolar stress induction, results in elevated protein levels of ATG7.

### 3.4. Interfering with RNA pol I Function by CX-5461 Stimulates Transcription of ATG7 and ATG16L1

Next, we impaired nucleolar function by use of a small-molecule drug to investigate if the effect on *ATG7* and other candidates may present a general mechanism of nucleolar stress induction. Thus, we used pharmacological inhibition of RNA pol I by application of the specific inhibitor CX-5461 as previously [25] to interfere with the very upstream event in ribosome biogenesis, transcription of the *47S rRNA* precursor, similar as after UBF-1 knockdown. Efficient induction of nucleolar stress was detected in HeLa and U2OS cells by qRT-PCR analysis for the *47S rRNA* precursor, which is expected to be decreased upon block of RNA pol I function (Figure 6A,C). As an independent approach to demonstrate efficient nucleolar stress induction, subcellular de-localization of the nucleolar factors NPM and the UBF-1 from nucleoli had earlier been demonstrated by immunohistochemistry in the utilized settings [25]. In HeLa cells exposed to CX-5461 we found that *ATG7* (Figure 6B) and *ATG16L1* expression was likewise upregulated, whereas the other candidates were not increased in this condition (Appendix A). We also found a significant, albeit weaker effect on *ATG7* in U2OS cells (Figure 6D), which correlated with decreased RNA pol I inhibition when compared to HeLa (compare Figure 6A,C). We also monitored protein levels of ATG7 in HeLa cells in response to CX-5461 and found a significant increase by Western blotting (Figure 6E,F). Protein levels of p53 were not stabilized in HeLa after CX-5461 treatment (Figure 6E,G). Consistently, the p53 target p21 was not increased, but rather decreased (Figure 6E,H), thereby suggesting p53-independent effects.

Given the increase of ATG7 and other autophagy factors exposed to nucleolar stress, we determined autophagic flux in response to CX-5461. Of note, CX-5461 has earlier been connected to activation of autophagy [11,24,32,45], however, flux studies have not been performed in detail, at least to our knowledge. We used well-established, stable HEK293A GFP-LC3 cells, which reflect a common model to monitor accumulation of GFP-positive autophagosomes and the state of lipidation of LC3 as readout for autophagy activation [30,40]. As a basis for our studies, we first verified functionality of CX-5461 in the cells by decrease in *47S rRNA* in presence and absence of the autophagy inhibitor chloroquine (CQ) (Figure 7A). Interestingly, we noticed that CQ treatment per se led to a decrease in *47S rRNA*. We also measured levels of *ATG7* (Figure 7B) and *ATG16L1* (Figure 7C) under these conditions. Whereas expression was not increased upon CX-5461 treatment in absence of CQ, we noticed a significant increase in response to CQ for all targets. The same was true for further autophagy factors *ATG5*, *ATG9A* and *ULK1*, with exception that *ATG4* was already induced in absence of CQ (Appendix A). Importantly, protein levels of lipidated GFP tagged LC3-II were significantly increased in flux assays with CQ in response to CX-5461 (Figure 7D,E) and correlated with increased ATG7 protein levels (Figure 7D,F), thereby suggesting activation of autophagic activity.

Quantification of GFP-LC3 positive puncta per cell by immunohistochemistry revealed accumulation of GFP-positive autophagosomes in CX-5461 treated samples. The number was increased in response to CQ in DMSO treated controls, in line with efficient block of autophagy (Figure 7G,H). We noticed a further increase in GFP puncta number per cell in response to CX-5461 in presence of CQ. Altogether the data show activation of autophagic flux in response to nucleolar stress by CX-5461, in line with previous reports.

In summary, knockdown of key ribosome biogenesis factors (PPAN, NPM, SBDS and PES1) or alternatively interference with RNA pol I function by depletion of the transcription factor UBF-1 or pharmacological inhibition by CX-5461 increased expression of the autophagy related genes *ATG7* and *ATG16 L1* in cancer cells. These findings strongly suggest that perturbation of ribosome biogenesis increases transcription of certain autophagy related factors as a general nucleolar stress response to possibly induce autophagy.

## 4. Discussion

The nucleolus has been established as critical stress sensor that reacts with a multitude of responses, such as p53 stabilization, cell cycle arrest, senescence, genome instability and apoptosis [3,46]. More recently, nucleolar stress has been found to function as upstream trigger for autophagy. Whereas several nucleolar factors and chemical compounds have been shown to participate in the regulation of autophagy, the underlying mechanisms have remained largely elusive. Involvement of p53-dependency and a contribution of mTOR signaling has been shown, at least under certain conditions [24]. 

In this study we provide novel evidence that depletion of key nucleolar factors previously linked to nucleolar stress induction (PPAN, NPM, SBDS and PES1) over-activate expression of the autophagy related genes *ATG7* and *ATG16L1*. The findings were recapitulated when interfering with the activity of RNA pol I by treatment with the specific small-molecule inhibitor CX-5461 [12] or when depleting the nucleolar RNA pol I transcription factor UBF-1. Given the independent approaches used, the data strongly argue for a transcriptional regulation of *ATG7* and *ATG16L1* in response to perturbation of ribosome biogenesis. Moreover, we also noticed upregulation of further *ATG* genes, such as *ATG4A*, *ATG5*, *ATG9* as well as *ULK1* after knockdown of nucleolar factors, all of which are required for early steps of autophagy and autophagosome biogenesis. 

We focused on ATG7 and showed accumulation of ATG7 on protein level in response to nucleolar stress in various cell lines. ATG7 resembles an essential E1-like enzyme being involved in initiation of the autophagic pathway to mediate formation of the autophagosomal membrane. With help of several other ATGs, ATG7 is essential to generate the lipidated form of LC3-I, the activated LC3-II that is in turn incorporated into the autophagosomal membrane. Whereas loss of ATG7 is well established to fully block autophagy [30], it was shown that enhanced expression of ATG7 per se is capable of increasing the autophagic flux [47]. Further studies in diverse models supported a connection of increased ATG7 levels, thereby positively correlating with LC3-II and autophagosome formation [48,49,50]. However, it should be noted that increased levels of ATG7 are currently under debate to serve as a reliable readout for activation of autophagy. Nevertheless, together with the increase in autophagic flux studies earlier observed in PPAN knockdown cells [34], the data presented here suggest an autophagic regulation that might already occur at transcriptional level. The same mechanistic principle might likewise account for effects previously observed on autophagy, for instance in cells exposed to CX-5461 or other nucleolar factors such as NPM and SBDS that have been linked to autophagy [24,31,32,33]. Indeed, we found enhanced levels of lipidated LC3-II in autophagy flux assays as well as increased number of GFP-positive autophagosomes after CX-5461 treatment. Interestingly, CQ treatment in HEK293A GFP-LC3 cells demonstrated synergistic effects by elevating transcript levels of *ATG7* and *ATG16L1*. This might well suggest that nucleolar stress activates autophagy already at transcriptional level by providing the factors required for early steps of autophagosome biogenesis. Note that NPM depletion, however, has earlier been shown to actually counteract autophagy activation in response to depletion of the RNA pol I transcription factor TIFIA in breast cancer cells [31]. Thus, different mechanisms might as well account for the effects observed in various settings, which should be further investigated in future.

Interestingly, also p53 is implicated in the regulation of autophagy at multiple levels. The *ATG7* promoter was previously shown to be bound by p53. Therefore, the effect on autophagy might as well depend on p53 and its transactivation of autophagy-related genes [51,52]. The induction of *ATG7* in this study was found in cancer cells that are considered both, p53-negative (HeLa) and p53-positive (U2OS). Hence, the effect seems to occur either irrespective of p53 status or at least different scenarios exist. In agreement with latter, effects on autophagy in response to nucleolar stress have also been found to occur both in p53-dependent and independent manner [24]. For instance, loss of NPM or SBDS has been shown to promote autophagy independently of p53, whereas CX-5461 was considered to function dependently and independently of functional p53 [11,24,31,32,33]. In principle, the family members p63 and p73 are capable of compensating for the loss of p53 with respect to *ATG7* expression [51,52]. Moreover, also in the p53-negative HeLa cell line an increase in p53 levels was detected in the context of PPAN depletion [38]. Thus, a dependency on the p53 family for the transcriptional regulation of *ATG7* cannot be fully excluded yet, at least for the conditions tested in this study. However, as we did not detect stabilization of p53 or its target p21 in response to CX-5461 in HeLa cells under the experimental setups used, the findings suggest p53-independent effects in this context. Besides p53, several unrelated transcription factors have been linked to the activation of *ATG7* expression [51]. Thus, further research is mandatory to unravel the precise mechanisms that drive the transcriptional activation of *ATG7* and further *ATG* genes in response to nucleolar stress.

Given the fact that nucleolar impairment by multiple triggers similarly induced expression of autophagy related factors, our findings suggest a novel nucleolar stress response that might globally be activated as compensatory mechanism to inhibit or delay apoptosis in the cells. Note that enhanced levels of ATG7, for instance, have been reported in diverse cancer types and were shown to promote cancer survival and resistance to chemotherapy, whereas a downregulation could counteract cancer development, progression and resistance [53,54]. Whether *ATG7* upregulation indeed represents a cause for diverse diseases associated with nucleolar stress needs to be further investigated. If so, we propose that interfering with ATG7 function, and with-it autophagy, might represent a novel strategy to consider in future in background of nucleolar-stress mediated malignancies. The same holds true for further autophagy regulators identified in this study. Moreover, autophagy inhibitors and activators should be tested in combination when applying novel anti-tumor strategies, such as CX-5461. 

## 5. Conclusions

Nucleolar stress has recently been linked to induction of autophagy [24]. The data provided in this study argue for a regulation that occurs already at transcriptional level. Together with our previous observations that nucleolar stress also over-activates Wnt target gene expression [25], our data add compelling evidence that the nucleolar stress response can function at RNA level to induce diverse pro-proliferative and pro-survival mechanisms. Depending on the context and intensity of the insult, nucleolar stress can either induce diverse pathogenic conditions linked to pro-proliferative diseases such as cancer or on the contrary to increased apoptosis as observed in neurodegeneration [15,24]. Likewise, nucleolar stress induction can be used as beneficial strategy to induce cell death in anti-cancer therapies in case of continuous and severe stress. Of note, over-activation of autophagy as well as inhibition are both capable of triggering apoptosis. Hence, an understanding of the multiple facettes of the nucleolar stress response will be beneficial to further optimize therapies underlying disturbed ribosome biogenesis.

## Figures and Tables

**Figure 1 cancers-13-06220-f001:**
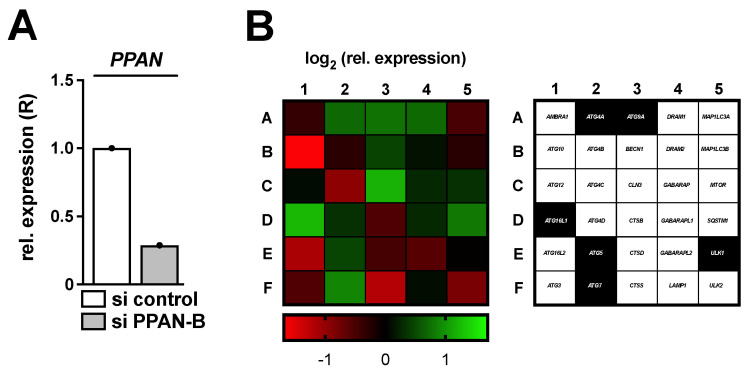
Knockdown of PPAN results in transcriptional alterations of autophagy related genes. HeLa cells were transfected with si PPAN-B for 48 h and subsequently analyzed by qRT-PCR. (**A**) Relative expression (R) of *PPAN* normalized to *GAPDH* is depicted. (**B**) The cDNA samples of (**A**) were subjected to a predesigned PrimePCR^TM^ Autophagy panel (BioRad) and the respective genes were normalized to *ACTB*, *GAPDH*, *HPRT1* as well as *TBP* (compare Appendix A). (**Left**) The normalized log_2_ relative expression upon PPAN knockdown in comparison to control transfected cells is depicted as a heatmap, (**right**) whereas right next to it the investigated genes are denoted. Thereby, the highlighted genes in black reflect targets, which were subsequently investigated.

**Figure 2 cancers-13-06220-f002:**
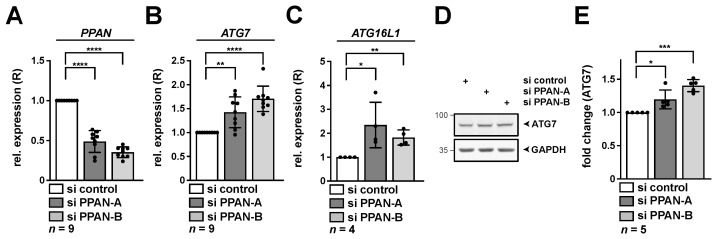
Depletion of PPAN leads to elevated amounts of *ATG7* and *ATG16L1* on mRNA and ATG7 on protein level. HeLa cells were transfected with PPAN siRNAs for 48 h as indicated and subsequently analyzed by qRT-PCR (**A**–**C**) and Western blotting (**D**,**E**). (**A**–**C**) The relative expression (R) of *PPAN* (**A**), *ATG7* (**B**) and *ATG16L1* (**C**) normalized to *GAPDH* is depicted. (**D**,**E**) RIPA whole cell lysates were subjected to Western blotting. Membranes were incubated with ATG7 and GAPDH antibodies. (**E**) Quantification of ATG7 normalized to GAPDH by densitometry as shown in (**D**). The si control was set to 1. Error bars represent S.D., *p* values were calculated by *t*-test. *, *p* < 0.05, **, *p* < 0.01, ***, *p* < 0.001, ****, *p* < 0.0001. *n* = number of independent experiments. The raw Western Blot can be found in Appendix A.

**Figure 3 cancers-13-06220-f003:**
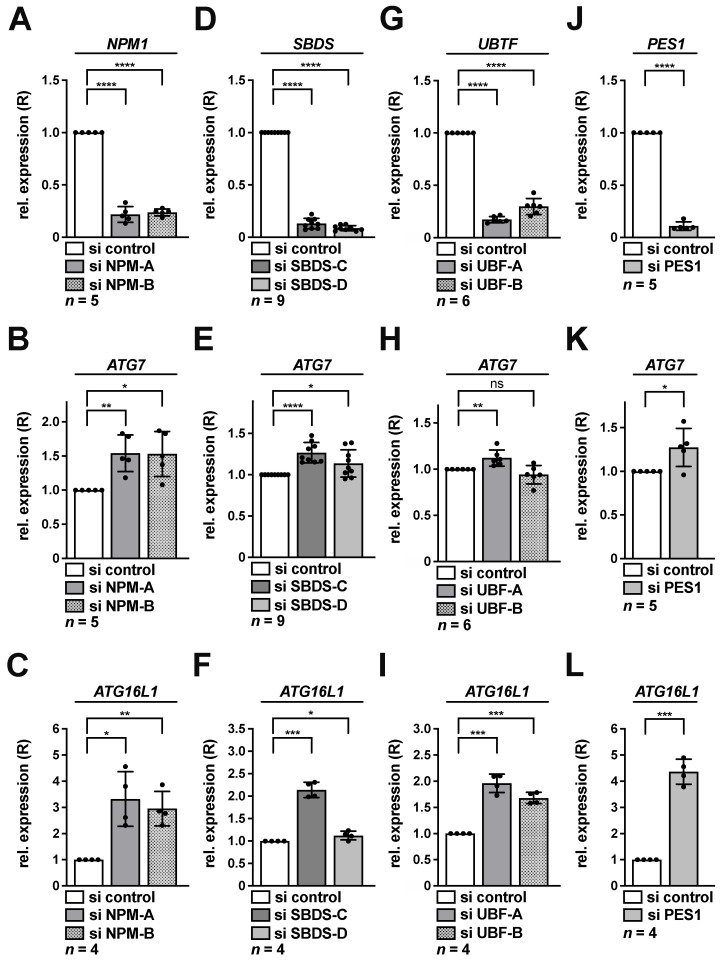
Knockdown of NPM, SBDS, UBF-1 or PES1 activates transcription of *ATG7* and *ATG16L1* in HeLa cells. HeLa cells were transfected with NPM (**A**–**C**), SBDS (**D**–**F**), UBF (**G**–**I**) or PES1 (**J**–**L**) si RNAs for 48 h as indicated and subsequently analyzed by qRT-PCR. Relative expression (R) of *NPM1* (**A**), *SBDS* (**D**), *UBTF* (**G**), PES1 (**J**), *ATG7* (**B**,**E**,**H**,**K**) or *ATG16L1* (**C**,**F**,**I**,**L**) normalized to *GAPDH* is shown. The si control was set to 1. Error bars represent S.D., *p* values were calculated by *t*-test. *, *p* < 0.05, **, *p* < 0.01, ***, *p* < 0.001, ****, *p* < 0.0001, ns, non-significant. *n* = number of independent experiments.

**Figure 4 cancers-13-06220-f004:**
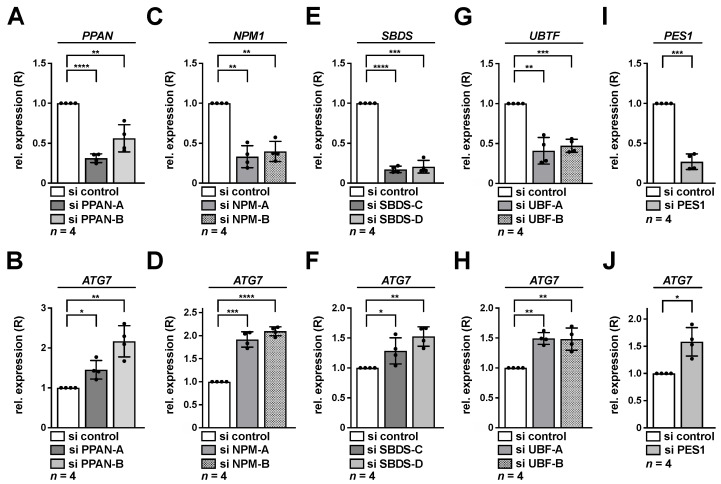
Knockdown of PPAN, NPM, SBDS, UBF-1 or PES1 activates transcription of *ATG7* in U2OS cells. U2OS cells were transfected with PPAN (**A**,**B**), NPM (**C**,**D**), SBDS (**E**,**F**), UBF (**G**,**H**) or PES1 (**I**,**J**) si RNAs for 48 h as indicated and subsequently analyzed by qRT-PCR. Relative expression (R) of *PPAN* (**A**), *NPM1* (**C**), *SBDS* (**E**), *UBTF* (**G**), PES1 (**I**) or *ATG7* (**B**,**D**,**F**,**H**,**J**) normalized to *GAPDH* is shown. The si control was set to 1. Error bars represent S.D., *p* values were calculated by *t*-test. *, *p* < 0.05, **, *p* < 0.01, ***, *p* < 0.001, ****, *p* < 0.0001. *n* = number of independent experiments.

**Figure 5 cancers-13-06220-f005:**
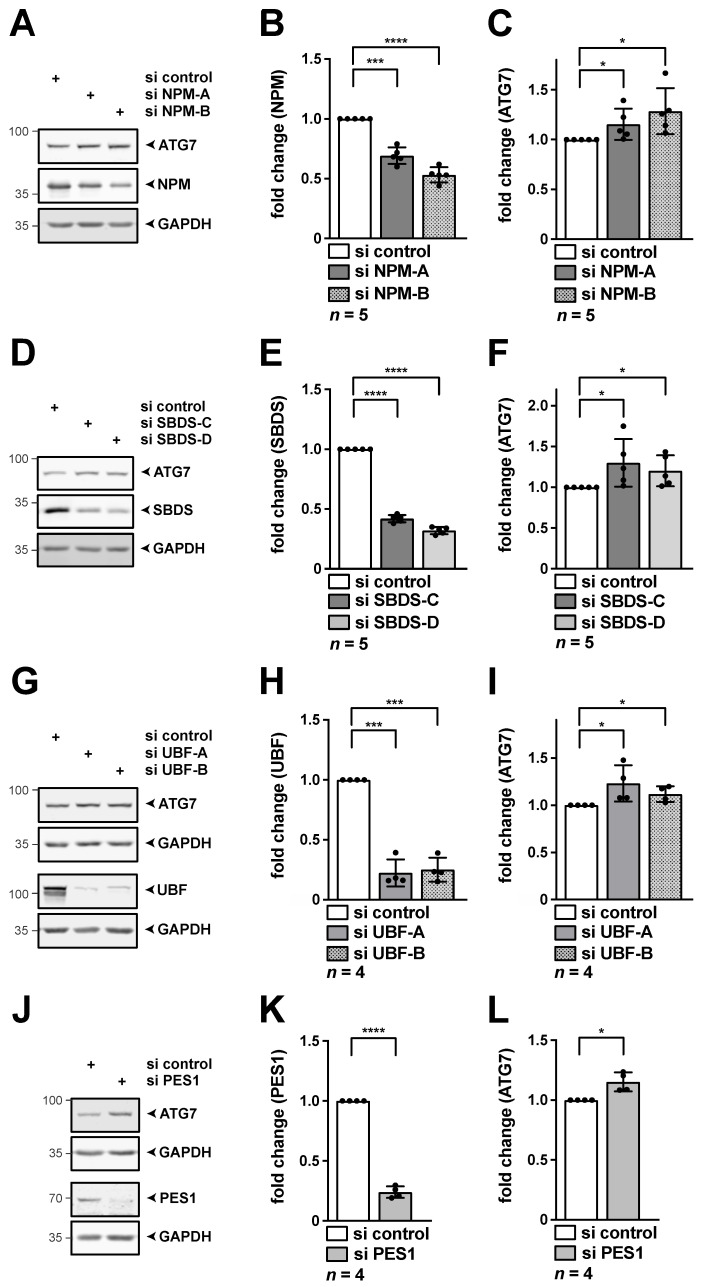
Knockdown of NPM, SBDS, UBF-1 or PES1 results in increased ATG7 protein levels. HeLa cells were transfected with the indicated si RNAs for 48 h. RIPA whole cell lysates were subjected to Western blotting, membranes were probed with NPM (**A**), SBDS (**D**), UBF (**G**), PES1 (**J**) as well as ATG7 and GAPDH antibodies as indicated. (**A**–**C**) Knockdown of NPM increases ATG7. Quantification of NPM (**B**) and ATG7 (**C**) normalized to GAPDH as shown in (**A**). (**D**–**F**) Knockdown of SBDS increases ATG7. Quantification of SBDS (**E**) and ATG7 (**F**) normalized to GAPDH as shown in (**D**). (**G**–**I**) Knockdown of UBF-1 increases ATG7. Quantification of UBF (**H**) and ATG7 (**I**) normalized to GAPDH as shown in (**G**). (**J**–**L**) Knockdown of PES1 increases ATG7. Quantification of PES1 (**K**) and ATG7 (**L**) normalized to GAPDH as shown in (**J**). Representative Western blots are shown, numbers indicate kDa. The si control was set to 1. Error bars represent S.D., *p* values were calculated by *t*-test. *, *p* < 0.05, ***, *p* < 0.001, ****, *p* < 0.0001. *n* = number of independent experiments. The raw Western Blots can be found in Appendix A.

**Figure 6 cancers-13-06220-f006:**
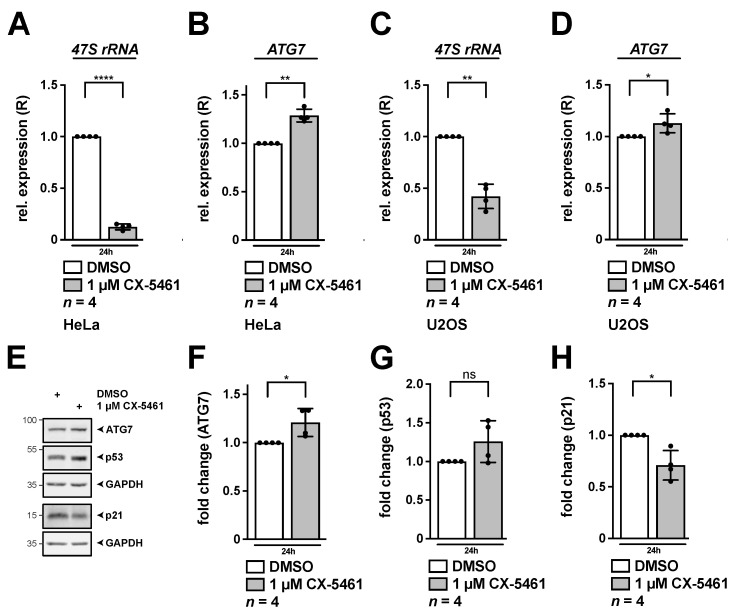
Inhibition of RNA pol I by CX-5461 leads to increased expression of *ATG7*. HeLa (**A**,**B**,**E**–**H**) or U2OS cells (**C**,**D**) were incubated with 1 µM CX-5461 for 24 h. (**A**–**D**) Relative expression (R) of *47S rRNA* (**A**,**C**) and *ATG7* (**B**,**D**) as determined by qRT-PCR normalized to *GAPDH* is shown. (**E**–**H**) RIPA whole cell lysates were subjected to Western blotting and membranes were incubated with ATG7, p53, p21 and GAPDH antibodies as indicated. Quantification of ATG7 (**F**), p53 (**G**) and p21 (**H**) normalized to GAPDH as shown in (**E**). DMSO treated cells served as control and were set to 1. Error bars represent S.D., *p* values were calculated by *t*-test. *, *p* < 0.05, **, *p* < 0.01, ****, *p* < 0.0001, ns, non-significant. *n* = number of independent experiments. The raw Western Blots can be found in Appendix A.

**Figure 7 cancers-13-06220-f007:**
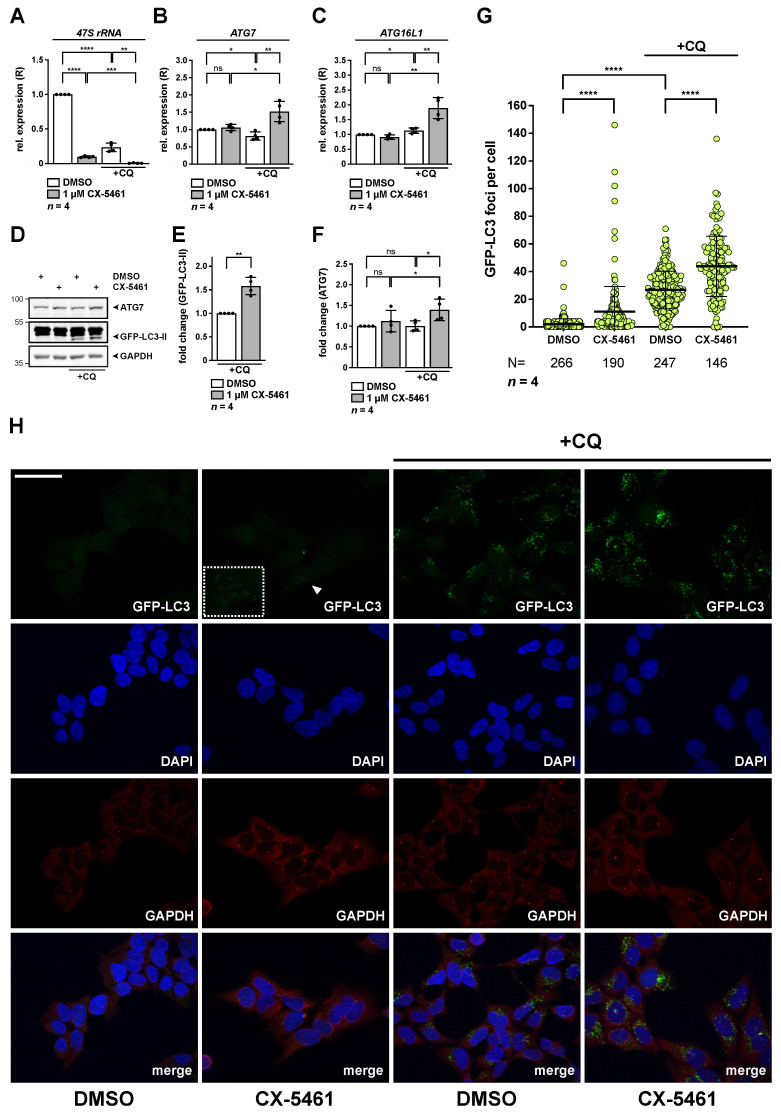
Induction of nucleolar stress by CX-5461 leads to increased expression of *ATG7* and enhances basal autophagic flux in HEK293A GFP-LC3 cells. HEK293A GFP-LC3 cells were incubated with 1 µM CX-5461 for 20.5 h and were subsequently exposed to chloroquine (CQ) for 3.5 h as indicated. (**A**–**C**) Relative expression (R) of *47S rRNA* (**A**), *ATG7* (**B**) and *ATG16L1* (**C**) as determined by qRT-PCR normalized to *GAPDH* is shown. DMSO treated cells served as control and were set to 1. Error bars represent S.D., *p* values were calculated by *t*-test. *, *p* < 0.05, **, *p* < 0.01, ***, *p* < 0.001, ****, *p* < 0.0001, ns, non-significant. *n* = number of independent experiments. (**D**–**F**) GFP-LC3-II protein levels increase upon treatment with CX-5461. RIPA whole cell lysates were subjected to Western blotting and membranes were probed with ATG7, GFP as well as GAPDH antibodies as indicated. Quantification of GFP-LC3-II upon CQ (**E**) and ATG7 (**F**) normalized to GAPDH as shown in (**D**). DMSO treated cells served as control and were set to 1. Error bars represent S.D., *p* values were calculated by *t*-test. *, *p* < 0.05, **, *p* < 0.01, ns, non-significant. *n* = number of independent experiments. (**G**,**H**) Inhibition of RNA polymerase I by CX-5461 increases accumulation of GFP-LC3 puncta in flux studies. Representative image stacks of randomly chosen optical fields are depicted in (**H**). Cells were stained with GAPDH antibody, nuclei with DAPI and GFP-LC3 was detected by fluorescence. The lower panels show the merged images of all channels. The arrowhead points towards representative GFP-LC3 foci in CX-5461 samples, which are magnified in the boxed inset. Scale bar, 50 µm. (**G**) Random image stacks as shown in (**H**) were counted for GFP-LC3 positive puncta per cell, indicating autophagosome accumulation. DMSO treated cells served as control. Error bars represent S.D., *p* values were calculated by unpaired *t*-test. ****, *p* < 0.0001. N = number of independently counted cells, *n* = number of independent experiments used for single cell counting. The raw Western Blot can be found in Appendix A.

**Table 1 cancers-13-06220-t001:** Overview of internal controls concerning the predesigned 96-well panel Autophagy (SAB Target List) H96 (BioRad).

	si Control	si PPAN-B
presence of gDNA	n.d.	n.d.
PCR inhibition (C_T_ < 30)	C_T_ = 22.9	C_T_ = 23.09
RNA degradation |RQ_2_ C_T_|− |RQ_1_ C_T_| < 3.0	RQ_1_ C_T_ = 19.33 RQ_2_ C_T_ = 20.52 =1.19	RQ_1_ C_T_ = 19.49 RQ_2_ C_T_ = 20.82 =1.33
RT control (C_T_ < 30)	C_T_ = 26.51	C_T_ = 27.07

## Data Availability

The data presented in this study are available in Appendix A.

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
