# Peer review of "Nucleolar Stress Functions Upstream to Stimulate Expression of Autophagy Regulators"

_cancers, 2021, doi:10.3390/cancers13246220_

Round 1
Reviewer 1 Report
The authors have adequately addressed most points we raised on the initial version.
Reviewer 2 Report
Dear Authors,
Thank you for addressing all points adequately. This increased the overall merit of your manuscript significantly.
Best regards
Reviewer 3 Report
Dear Editor,
I congratulate the authors for the excellent work. I believe that the added data have helped to broaden and strengthen the results obtained.
Best regards
This manuscript is a resubmission of an earlier submission. The following is a list of the peer review reports and author responses from that submission.
Round 1
Reviewer 1 Report
The authors reported that nucleolar stress could stimulate the expression of the autophagy regulator ATG7. It's a nice study. However, this paper needs substantial revision.
1] Author should measure the western blot intensities that are not seemed matching.
2] Author should show data of autophagosome production in cells by staining, ICC, and LC3 localization.
3] Did the authors checked other ATGs? should show other ATGs expressions.
4] Authors should confirm the siRNA results by overexpression of ribosome biogenesis factors. For example, PPAN, NPM1, SBDS, and PES1.
5] In Fig. 5, what happened if PPAN or other factors overexpressed in CX-5461 treated cells? Authors should show this data.
Author Response
see word file attached

Reviewer 2 Report
It has been shown that autophagy is induced upon the induction of nucleolar stress, which is considered to be a cytoprotective mechanism. This article deals with the question how nucleolar stress explicitly induces the formation of autophagosomes. Here, nucleolar stress is induced by depletion of nucleolar resident early ribosome biogenesis factors NPM1, PPAN, UBF-1 and Pes1, late ribosome maturation factor SBDS as well as by using the Pol-I inhibitor CX-5461. The authors describe the upregulation of the autophagy factor ATG7, which is essential for the conjugation of ATG8/LC3 proteins to the autophagosomal membrane during autophagosome maturation. By predominantly measuring mRNA levels by qPCR they claim this effect relies on an effect of nucleolar stress on the transcriptional upregulation of ATG7 mRNA. Moreover, using HeLa (considered p53 negative) and U2OS (p53 positive) cells, they discuss the involvement of the important transcription factor p53 in this process. My main concern is that this work reveals only very minor changes in ATG7 expression upon induction of nucleolar stress. Therefore, at least to my opinion, the physiological relevance of the finding remains vague and without further data the link from nucleolar stress to enhanced ATG7 expression resulting in autophagy induction is rather speculative.
Major points:
-The key message of this report is that that the enhanced autopagosomed formation and autophagic flux upon nucleolar stress is due to the induction of ATG7 expression. Given the very minor changes in ATG7 expression shown here it would be important to confirm enhanced autopagosomed formation and autophagic flux under the respective experimental conditions (e.g. depletion of PPAN, NPM1, SBDS1).
-The authors state that they did a qRT-PCR screen, in which they identified the induction of ATG7 expression upon PPAN depletion but do not show the data. They should include this experiment in the publication.
-Along this line, the expression of proteins involved in autophagy is regulated in a coordinated and orchestrated way. Some examples are shown in Füllgrabe et. al. The authors should test the effect of nucleolar stress on mRNA and proteinlevels of other proteins involved in autophagy such as Ulk1, ATG5 and LC3/GABARAPs.
-The effects on ATG7 upregulation are relatively weak especially on protein levels. In order to strengthen the effect of nucleolar stress on the transcription of autophagic factors, also other factors should be tested (see above).
-The authors discuss that even though HeLa cells have an increased degradation rate of p53 (due to the expression of E6 oncoprotein), p53 proteins was detected in HeLa cells upon PPAN depletion before. Since the stabilization of p53 is one of the hallmarks of nucleolar stress, its protein levels should also be checked with the other conditions and correlated to the effect on ATG7 expression. This is especially important because ATG7 was already described to be under the control of p53 transcription factor (Füllgrabe et al.). In this line it would be important to use p53 negative cell line, such as SAOS2, or p53 ko cells to investigate the role of p53 in ATG7 upregulation.
Additional points:
-page 2 line 91: ATG7 is not an E1-ubiquitin ligase but an E1-like enzyme and should be rephrased
-to Fig. 5: It might be that U2OS cells have a higher tolerance towards CX-5461 treatment. Higher concentration of this inhibitor might result in stronger induction of nucleolar stress and thereby to a stronger effect on ATG7.
-For consistency, ATG7 protein levels should also be shown upon CX-5461 treatment and UBF-1 depletion.
-For consistency, ATG7 mRNA levels upon UBF-1 depletion should also be shown in HeLa cells.
-Figure 3 and 4 (same experiments but different cell lines) should be combined or parts subjected to supplementary figures.
-The authors should discuss the controversy that NPM1 depletion promotes autophagy by inducing nucleolar stress, while it is also necessary to induce autophagy upon induction of nucleolar stress (Katagiri 2015).
-In general it would strengthen the publication not only to refer to other publications in which increased LC3 lipidation was shown upon induction of nucleolar stress, but include LC3-lipidation assays in this study as well using the same conditions.
-The authors should consider to use also other normalization factors than GAPDH because this might lead to a strong bias.
Author Response
see word file attached

Reviewer 3 Report
Dear Authors,
Your manuscript on the regulation of ATG7 upon nucleolar stress is a valuable contribution to autophagy research. It is very well written and I especially value the comprehensive introduction to this topic. Nevertheless I do have some minor points that you could consider to improve the manuscript.
- The scale bars in the figures are not always comparable. For the mRNA expression only Figure 4 C and 5 B is different form the others, but for the western blot quantification the scale bars vary from 1,5 up to 3,0 as a miximum across all figures. I would recommend to adjust them to one maximum within one figure to allow better comparison of the data. Best would be, if they could be adjusted to the scale bar of the mRNA, so that one can easily see the effect of mRNA abundance has on protein level.
- The information given in line 347 on the p53 status of the cell lines would be good to know upfront. This would make it easier to understand the reasons why these two different cancer cell lines were chosen. I would recommend to put this information in 3.1.
- Could you please explain why you only used U2OS cell for the last experiment (Figure 6)? Even if HeLa experiments did not result in any major response it would be worth showing this data.
Best regards
Author Response
see word file attached

Reviewer 4 Report
The ribosomal biogenesis is a very important process for the maintenance of cellular turnover, which takes place in the nucleoli. Nucleolar stress often causes alterations in the ribosomal biogenesis, causing a series of serious pathologies. Therefore, knowing the mechanisms that cells activate in response to nucleolar stress is an extremely important goal. On the other hand, as the authors themselves write, the link between nucleolar stress and autophagy is already known. In my opinion, therefore this work is not very innovative, reiterating, in fact, that nucleolar stress activates an autophagic process.
However, the topic is of great scientific interest, the experimental design is well conducted, the results of the experiments performed support the conclusions and the paper has been clearly written.
Author Response
see word file attached

Reviewer 5 Report
Dannheisig and co-workers are interested in the mechanism by which autophagy is induced following nucleolar stress. The authors show that upon induction of nucleolar stress by knocking down several genes, ATG7 is slightly upregulated. The manuscript is well-written, with an informative introduction, clear description of performed experiments and well-informed discussion. The methods are also described in sufficient detail. The manuscript could however benefit from further verification of the mRNA data in some experiments. It also appears a bit meagre, as all data could be summarised in one image: knockdown of several factors by siRNA or inhibition of RNA pol 1 lead to a slight increase in ATG7 mRNA and potentially protein expression. Whether this is sufficient to actually stimulate autophagy is not clear as the authors rightly indicate in the discussion.
- It is not clear how the initial qRT-PCR screen for autophagy targets was performed. If this screen were to be included, it would greatly improve the manuscript.
- The current figures rely heavily on RT-qPCR data. The manuscript would improve tremendously if all experiments were confirmed by western blot. It will be interesting to see for example to what extent the approximately 50% knockdown of PPAN on mRNA level leads to loss of protein. If PPAN has a long half-life, then there could the substantial protein amounts left.
- The effects on ATG7 protein level in 1E/4C/4F seem rather small. Do the authors expect that this slight increase in ATG7 will have a major role in stimulating autophagy upon nucleolar stress? It would be helpful to indicate the ranges wherein other publications have reported that ATG7 is upregulated to stimulate autophagy.
- The western blotting data of figure 4 should be merged with the qRT-PCR data presented in figure 3. The knockdown efficiency on RNA level in figure 3 becomes redundant with the efficiency measured using western blotting in figure 4, as the protein levels are the most relevant parameter.
- In figures 5 and 6 western blots of ATG7 should be included, to determine whether the slight increase on mRNA level also here corresponds with an increase on protein level. As it stands, one wonders why it is not included - was it simply not done, or was no effect seen?
- The conclusion “our data establish that nucleolar stress affects transcriptional regulation of autophagy” is too strong. The only thing these data show, is that ATG7 is slightly upregulated. To be able to make such a strong conclusion, the authors would have to measure autophagy in these conditions and next analyse whether any measured effect is mediated by the enhanced levels of ATG7. As it stands, the only conclusion that can be drawn is that certain nucleolar stresses lead to a (slight) increase in ATG7 levels. I would recommend to either down-tune this conclusion or to include more data that support it.
Author Response
see word file attached
